# Study on Bone-like Microstructure Design of Carbon Nanofibers/Polyurethane Composites with Excellent Impact Resistance

**DOI:** 10.3390/nano12213830

**Published:** 2022-10-29

**Authors:** Jun Gao, Hongyan Yang, Zehui Xiang, Biao Zhang, Xiaoping Ouyang, Fugang Qi, Nie Zhao

**Affiliations:** 1School of Materials Science and Engineering, Xiangtan University, Xiangtan 411105, China; 2Key Laboratory of Low Dimensional Materials and Application Technology of Ministry of Education, Xiangtan University, Xiangtan 411105, China

**Keywords:** impact resistance, bone-like structures, carbon nanofibers, energy absorption

## Abstract

It is a challenge to develop cost-effective strategy and design specific microstructures for fabricating polymer-based impact-resistance materials. Human shin bones require impact resistance and energy absorption mechanisms in the case of rapid movement. The shin bones are exciting biological materials that contain concentric circle structures called Haversian structures, which are made up of nanofibrils and collagen. The “soft and hard” structures are beneficial for dynamic impact resistance. Inspired by the excellent impact resistance of human shin bones, we prepared a sort of polyurethane elastomers (PUE) composites incorporated with rigid carbon nanofibers (CNFs) modified by elastic mussel adhesion proteins. CNFs and mussel adhesion proteins formed bone-like microstructures, where the rigid CNFs are served as the bone fibrils, and the flexible mussel adhesion proteins are regarded as collagen. The special structures, which are combined of hard and soft, have a positive dispersion and compatibility in PUE matrix, which can prevent cracks propagation by bridging effect or inducing the crack deflection. These PUE composites showed up to 112.26% higher impact absorbed energy and 198.43% greater dynamic impact strength when compared with the neat PUE. These findings have great implications for the design of composite parts for aerospace, army vehicles, and human protection.

## 1. Introduction

During the last decades, the dynamic behaviors of composite materials have been emerging as the focus area of researchers due to complex and volatile international relations. Optimal and practical design of lightweight and high-performance impact-resistant structures is crucial for human safety, defense industry, and aerospace [1,2,3]. The high-speed impact often occurs when high-speed and high-energy shot/debris directly impact on the target, which means the flexible protective devices have excellent strength and high resilience, which can avoid penetrating damage and absorb impact energy [4,5]. With the development of advanced nanofillers and synthesizing techniques, polymeric nanocomposites have become a great level of materials of dynamic impact resistance [6,7,8,9]. Therein, polyurethane elastomers (PUE), a kind of flexible, superior impact-induced stiffening, and strengthening polymer, have become one of the most frequently applied materials in dynamic impact events [9,10,11]. Compared with traditional polymer-reinforced materials, carbon nanofibers (CNFs) have the advantages of high strength, energy dissipation, and excellent thermal stability under extreme working potential [12,13]. However, it is not a good strategy to add CNFs directly to PUE, which causes the agglomeration of CNFs in the matrix and decreases the mechanical property of polymer composites [14,15]. Up to now, it has been a great challenge to design cost-effective and impact-resistant polyurethane composites with specific microstructures.

Many researchers have turned to natural biology to find optimal structures and creatively design different structures to improve their impact resistance. These impact-resistant biomaterials mainly contain the following characteristics: porous, multilayer, gradient, composite, interface, viscoelasticity, and viscoplasticity [1,16,17,18,19,20,21]. Li et al. [19] studied how hedgehog spin worked to keep the balance between stiffness and toughness and found that these spines have a sandwich structure with an inner porous core combined with an outer dense cortex, which improves impact resistance and structural stability of composites. Zhang et al. [20] designed a nacreous structure by Al_2_O_3_-rGO/Al laminated board inspired by nacre with simultaneous enhancement of the modulus-strength-ductility. Wang et al. [21] prepared a kind of glass fiber-reinforced laminated board mixed with carbon nanotube-modified epoxy and pre-stretched fiber fabrics with excellent dynamic thermo-mechanical properties and superior impact resistance. In fact, those biological materials have excellent impact resistance due to the synergy of multiple structures rather than a single structure, leading to impressive properties under dynamic loading.

Human shin bones, which contain many characteristics of impact-resistant materials, are an impact-resistant structural material with a combination of soft and hard, and they are made up of hollow mineral fibrils and organic proteins on the nanoscale [22,23]. The regular arrangement of mineral fibrils contributes to bone strength, and these 85–90% of proteins are composed of collagen, which play a great role in bone flexibility and energy absorption [22,24,25]. Many studies focused on the human bone structures and found that there are many concentric circle structures, which are called Haversian structures, and demonstrated that these concentric microstructures with soft and hard materials exhibit a better balance between strength and toughness, as shown in Figure 1a [17,26,27]. In addition, CNFs are widely used for bone tissue engineering applications because of their biocompatibility and splendid theoretical mechanical properties with nanoscale fiber similar to crystalline hydroxyapatite, which is found in physiological bone, suggesting that it is a strong possibility for use as an orthopedic implant material [28,29].

Furthermore, natural mussels can firmly adhere to smooth stones to resist choppy waves. It has been that the mussel foot-byssal proteins (Mfps) have superior adhesion capacity and toughness, and they are widely used in the surface modification of nanofillers [30,31,32]. Many studies on mussel adhesion proteins modified nanofillers have focused on polydopamine (PDA). However, Fewer studies on materials for cheaper and more efficient synthesis of mussel adhesion proteins. Besides, some studies found that the secret of adhesion behaviors are from Mfps, in which the catechol and amido play an essential role in adhesive behaviors [33,34]. What is more, collagen-mediated adhesion between inert surfaces is weak, Nadine et al. [35] used mussel foot protein–3 (Mfp–3) to provide a cohesive matrix for collage, which suggests the potential of Mfps for biomedical applications.

In view of the excellent impact resistance of human shin bones and the potential of CNFs and mussel adhesive proteins as orthopedic implant materials, we pay more attention to designing the biological structure of the shin bone to achieve better mechanical properties and impact resistance of polymer matrix composites. Therefore, we made an exciting study on the adhesion chemistry of mussels and bone-like microstructural materials with impact resistance. Inspired by mussel adhesive proteins in mussels, we creatively chose catechol (CC) and polyethyleneimine (PEI) to bionic design mussel adhesion proteins to improve the interfacial bonding and compatibility between CNFs and PUE, as shown in Figure 1b. Besides, CNFs were regarded as tiny bone structures, and we used cheaper CC and PEI to co-deposit on the CNFs, serving as sticky collagen, which provided unexceptionable toughness and adhesive power. What is more, the CC-PEI made CNFs functionalized so that they could disperse evenly in PUE, and PUE composites with bone-like microstructures, had excellent impact resistance as shown in Figure 1c. We conducted the static compression experiment, split Hopkinson bar (SHPB) test, and dynamic thermomechanical analysis (DMA) test to measure the static/dynamic and interface problems of PUE composites, and data exhibited that PUE with bone-like microstructures could significantly increase the energy absorption, dynamic strength, compression resilience rate, and storage modulus of PUE composites.

## 2. Material and Experimental Methods

### 2.1. Materials

PUE raw materials, which are composed of A component (2,4-TDI, –NCO, 6.21 wt.%) and B component (poly(oxycarbonyloxy-1,6-hexanediyl), –OH, 6.36 wt.%), were provided by Qingdao Green World New Material Technology Co., Ltd (Qingdao, China). Carbon nanofibers (CNFs, ≥97%, diameter: 50–200 nm, length: 1–15 μm) were purchased from Jiangsu XFNANO Materials Tech Co., Ltd. (Nanjing, China), other chemicals including catechol (CC, ≥99.0%), polyethyleneimine (PEI, M = 600, ≥99.0%), Tris-HCl (≥99.0%), hydrochloric acid (HCl, AR, 36.0~38.0%) were purchased from Aladdin.

### 2.2. Preparation of CC-PEI @CNFs

As shown in Figure 1b, 0.1 g Tris-HCl and 200 mL H_2_O were added to 500 mL flask, the PH was adjusted to 8.5 with HCl. Then 1.0 g CC, 0.5 g PEI and 0.7 g CNFs were added into a flask and stirred at 25 °C for 36 h. Then, they were washed several times with water to neutral, filtered, and dried at 60 °C for 24 h. These modified CNFs were named CC-PEI@CNFs. The mass ratio of CC to PEI was at 2:1, and we changed the concentration of CC and PEI, as shown in Table 1, which was named CPC–1, CPC–2, and CPC–3, respectively, and all other experimental conditions remained the same.

### 2.3. Preparation of CC-PEI@CNFs/PUE (CPC/PUE)

CC-PEI@CNFs, A and B components were added into the beaker and stirred for 20 min. Then, they were put into a vacuum deaeration machine to pump air bubbles for 5 min, and finally poured into the mold and cured, lasting 3 days. The mass ratio of component A to component B was maintained at 5:2, and the mass rate of CPC–1, CPC–2, CPC-3, and CNFs were maintained at 1.0% of the entire mass of A and B components. These nanofillers were added into A and B components, named CPC/PUE–1, CPC/PUE–2, CPC/PUE–3, and CNFs/PUE, respectively. It was considered that CPC/PUE-2 had the best dispersion in ethanol, as shown in Appendix A, which is beneficial to improve the mechanical properties of polymer composites. Thus, we continued to study the effect of mass fraction on CPC/PUE–2, and the mass fraction of nanofillers was 0.2%, 0.4%, 0.6%, 0.8%, 1.0%, and 1.2% of the entire mass of A and B components, named CPC/PUE–Ⅰ, CPC/PUE–Ⅱ, CPC/PUE–Ⅲ, CPC/PUE–Ⅳ, CPC/PUE–Ⅴ, and CPC/PUE–Ⅵ, respectively, as shown in Table 2.

### 2.4. Characterizations

Fourier transform infrared (FTIR) spectrum was obtained by a Thermo Scientific Nicolet iS5 (Waltham, MA, USA). The polymerization of CNFs and the thermal stability of elastomers were detected by a thermal analyzer (TGA, TGA5500, New Castle, DE, USA) in N_2_ atmosphere at a heating rate of 10 °C/min. Absorption peak of aqueous solution of nanofillers was measured with an Ultraviolet spectrophotometer (UV-Vis, Agilent, CARY100, New Castle, DE, USA) from 700 nm to 250 nm. X-ray photoelectron spectroscopy (XPS, Thermo Scientific K-Alpha, Palo Alto, CA, USA) was used to characterize the surface chemical components. An optical microscope (Olympus, BX53M, Tokyo, Japan) was used to observe the dispersity of nanoparticles in ethanol solution. Scanning electron microscopy (SEM, Zeiss, Sigma300, Baden-Württemberg, Germany) was devoted to observing the distribution status of CC-PEI@CNFs surface and fracture surface morphology of PUE composites. Transmission Electron Microscope (TEM, TF20, Palo Alto, CA, USA) was used to observe the surface morphology of CNFs and CC-PEI@CNFs.

### 2.5. Quasi-Static Compression Test

Quasi-static compression tests were carried out on the universal electronic testing machine (Hua Long, WDW-100C, Changsha, China). The sample size was Φ20×4 mm cylinder. The test parameters referred to ISO 7743 2008(E) and were adjusted according to the actual experimental situation. The loading rate of the tests was 1.2 mm/min, and the compression displacement was 3.5 mm. All samples were subjected to three parallel experiments and averaged.

### 2.6. Dynamic Mechanical Analysis Test

DMA (Eplexor500N, Koenig & Bauer AG, Baden-Württemberg, Germany) was used to study the dynamic mechanical properties and the interface problems of viscoelastic and viscoplastic solid. DMA test conditions were as follows: Test sample size: Long strip 35 × 5 × 2 mm, tensile strain: 1.0%, dynamic strain: 0.2%, frequency: 5 Hz, test temperature range: −60 °C–70 °C. All samples were subjected to three parallel experiments and averaged.

### 2.7. SHPB Experiment

The SHPB experiments were set up to measure the dynamic mechanical response between the impacting bar and the target samples. The schematic diagram of the composition of the SHPB device is shown in Figure 2a, which is composed of 4 bars (bullet, incident bar, transmission bar, and absorber bar), which are made of aluminum alloy (elastic modulus E = 70 GPa, Poisson’s ratio υ = 0.33, density ρ = 2.8 g/cm^3^, diameter Φ = 20 mm). The working of the device is based on the theory of one-dimensional wave propagation in elastic bars. In Figure 2b, after adjusting the instrument, the interface strain time curve of the incident bar and sample (incident wave + reflected wave) coincides with the interface strain time curve of the transmitted bar and sample (transmitted wave), indicating that the samples reach a good stress balance during loading, which verifies the validity of the experiment [36]. The size of cylindrical specimens was Φ10×2 mm. A smaller aspect radio (L/D = 0.2) is helpful for soft materials like PUE to avoid the end effects and inertia effects and maintain uniform deformation and stress balance [37,38]. In view of the PUE exhibiting strain rate dependency of its mechanical properties, it is necessary to control the same strain rate by controlling the gas pressure parameters of the apparatus at room temperature (20 °C) to measure the effect on different mass fraction nanofillers into PUE [38,39,40]. The results show that strain rates were controlled at ~9000 s^−1^ (Figure 2c). In all experiments, petroleum jelly lubricant was used at the interfaces between the bar and sample to minimize the interfacial friction. All samples were subjected to three parallel experiments and averaged.

## 3. Results and Discussion

### 3.1. Structure Characterizations

FTIR, XRD, UV-Vis, XPS, and TGA were used to prove the successful synthesis of CC-PEI on the CNFs. From the FTIR spectra in Figure 3a, it can be seen that new absorption peaks of N-H (1540 cm^−1^), C-N (1350 cm^−1^), and C-O (1110 cm^−1^). Appendix A shows that XRD pattern consisted of a broad diffraction peak at 20.72° without any detectable sharp peaks, which indicates the amorphous molecular structure of samples. As is shown in Figure 3b, a new signal, N1s at 400.1 eV, [41,42,43] appears in the spectra of CC-PEI@CNFs, and N1s accounts for 3.35% of CPC-2. Figure 3c shows three broad peaks at around 280, 322, and 480 nm in the solution, which is suggested as evidence for the cross-linking reaction [41,42]. Appendix A shows the N1s pattern of CNFs and CC-PEI@CNFs, and XPS N1s spectrum of CC-PEI@CNFs can be curve-fitted with three peak components at a binding energy of 398.6, 399.9, and 401.4 eV, which correspond to –NH_2_, –R_2_NH, and –N=, [43,44] respectively. Besides, for CNFs, the XPS C1s spectrum can be curve-fitted with three peak components at a binding energy of 284.8, 286.6, and 289.0 eV, attributable to the C-C, C-O, and O=C-O species, respectively (Figure 3d). However, for CC-PEI@CNFs, the C1s spectra appear at a new peak at 285.5 eV, which represents C-N (Figure 3e), which is suggested as evidence for the cross-linking reaction [41,42]. From Figure 3f, we find that CNFs lose about 6% weight at 800 °C, due to the loss of water and oxide decomposition. As for the different concentrations of CC-PEI to modify CNFs, the weight of CC-PEI@CNFs decreases with the temperature rising, and high concentrations of CC-PEI lose more weight. TGA spectra of CC-PEI@CNFs can be divided into three stages. Take CPC-1, for instance: 30–220 °C, nanofillers lose weight by almost 4% on account of absorbed water, 220–580 °C, they lose about 14% due to the decomposition of CC-PEI on the CNFs. 580–800 °C, they lose about 3% due to C-O and O=C-O decomposed on the CNFs. Therefore, the weight loss of CC-PEI@CNFs at different concentrations is 17%, 33%, and 34%, respectively. The above experiments could prove the success in synthesizing CC-PEI@CNFs.

Microscopic morphology of composite nanoparticles is observed by optical microscopy and SEM. In comparison to the agglomeration of CNFs, the addition of CC-PEI on the CNFs can observably improve the dispersity of nanofillers in the ethanol solution. Therein, CPC–2 has the best dispersity property, as shown in Appendix A. From Figure 4, we can clearly see the morphology of CNFs and CPC–2. CNFs have the interstitial hole from the TEM images (Figure 4a_4_,b_4_). Contrary to the smooth surface of CNFs, the surface of CPC–2 is rougher and thicker. The thickness of the coating on the surface of CPC–2 is about 15 nm, with a relatively even and complete package, and nitrogen element of CPC–2 is about 8.1% from SEM line scanning test, as shown in Table 3. Appendix A shows the SEM map scanning of CNFs and CPC–2.

### 3.2. Mechanical Properties of PUE Composites

#### 3.2.1. Quasi-Static Compression Performance

Considering that the elastomers display significant linear deformation and strain hardening throughout the compression process, it is necessary to compute the compression modules, compression resilience rate, and static energy absorption [37,38], which are on behalf of the strength and toughness of PUE composites. The corresponding data is shown in Figure 5. Typical linear elastic strain as a function of stress can be observed in the initial stage (Figure 5e). Then, the curves rise steeply with the further increased strain, which is caused by the strain hardening effect. After compression, CPC/PUE do not exhibit significant macroscopic deformation (Figure 5d), while the neat PUE and CNFs/PUE exhibit severely irreversible deformation and failure (Figure 5c). From Figure 5e, the strain-stress curves have excellent repeatability. Leaving the mass fraction of nanofillers 1.0 wt.%, CC-PEI are set as three different concentrations and CPC/PUE–2 have the best mechanical properties. We also designed different mass fractions of CPC at the base on the CPC/PUE–2 to find the relationship of mechanical property influence. The data show that, with the increase of CC-PEI@CNFs, the compression modulus, compression resilience rate, and static energy absorption of composite PUE have a first rise and then fall, therein 1.0 wt.% CPC/PUE has the best static compression properties: Compression modulus is increased by 149.08% (52.53 ± 0.25 MPa), compression resilience rate is 94.7%, and the absorption energy capacity of CPC/PUE is increased by 84.35% when are compared with neat PUE. In comparison, CNF/PUE has a bad toughness and only slight promotion on strength.

#### 3.2.2. DMA of PUE Composites

Intensive elastic-plastic deformation behaviors are observed in the PUE samples, which are beneficial to improve the impact resistance of the materials [17,45,46]. DMA is the most extensive and practical method to investigate the mechanical and solid viscoelastic properties of composites over a wide temperature range.

The glass transition temperature (T_g_) is usually taken as the corresponding temperature of tan δ peak, reflecting the mobility of polymer segments during the heating process [38,47]. The T_g_ of neat PUE was about −35.6 °C. The variations in storage modulus (E’) and dissipation factor (Tan δ) with temperature, as shown in Table 4 and Figure 6.

After adding CC-PEI@CNFs fillers, the storage modulus of the polymer significantly increases and shows obvious regularity, i.e., with the increase of CC-PEI@CNFs, the E’ of the PUE composites increases first and then decreases in the glass temperature (<T_g_) and high elastic temperature (>T_g_) range as shown in Figure 6a and Table 4. While the T_g_ and the T_g_ peak value slightly decreases too. Therein, the maximum peaks of E’ near the glass region (−60 °C) and the rubber region (20 °C) are 1293.42 ± 69.23 MPa for CPC/PUE–III and 55.29 ± 1.09 MPa for CPC/PUE–V, which are 54.50% and 43.72% higher than the neat PUE, respectively.

For one thing, the enhancement effect and internal friction of the nanofillers lead to the improvement of the E’ of PUE composites. For another, the CC-PEI@CNFs can reduce the T_g_, showing that the modified nanofillers enhance the mobility and elasticity of PUE chain in the polymer composites. While the CNFs/PUE show higher E’ and T_g_ (Figure 6c,d), exhibiting the fact that CNFs only enhance the strength of PUE composites but not their toughness. The strengthening and toughening effect can significantly improve the mechanical properties of PUE and meet the requirements of impact-resistant materials.

The data show that the PUE containing CC-PEI@CNFs present obviously higher E’ and lower T_g_ for the entire temperature range, which is attributed to the smaller cluster size and the resultant larger interface area. In addition, the bone-like structures show good modulus and toughness and play a powerful supporting role in the PUE matrix. The surface of CC-PEI@CNFs contains abundant hydrogen bonds, π-π bonds, and other unsaturated bonds, as well as strong covalent bonds like hydrogen bonds, –NH/NH_2_. The addition of fillers improves the strength of the polymer and increases the flexibility of movement between chain segments.

Obviously, the above results confirm that incorporating a small quantity of modified CNFs into PUE can balance strength and toughness, and these tests are favorable to the further improvement of the impact property.

#### 3.2.3. High Strain Rate Compression Test

The SHPB is a device capable of applying a strain rate close to that imposed by a ballistic impact of a pistol, which is the minimum performance requirement for ballistic resistance for military equipment [40,48]. The energy absorption, which is used to reflect the impact-resistance capacity of the specimens, is defined as the area under the stress-strain curve, and a larger area implies a higher energy absorption property and also superior impact-resistant properties [49]. The results indicate that the addition of CC-PEI@CNFs can affect the dynamic strength, elongation, and absorbed energy of the PUE composites. When the additional amount of CC-PEI@CNFs reaches 1.0 wt.%, the dynamic strength of CPC/PUE comes up to 374.3 ± 15.3 MPa, the elongation comes up to 1.42, and is enhanced by about 198.43% when compared with the neat PUE. Energy absorption capacity of CPC/PUE–Ⅴ is enhanced by about 112.26%, and their surface shows tiny radial and axial crazes (Figure 7f), while the CNFs/PUE show lower elongation and complete penetration behaviors, as shown in Figure 7 and Appendix A.

The post-test observation of PUE material by SEM reveals the internal mechanism of crack behavior and dynamic strain energy absorption and impact resistance of PUE composites. The SEM morphologies of the PUE sectional view of PUE composites are shown in Figure 8 and Appendix A. after SHPB test via liquid nitrogen brittle fracture.

The mechanical enhancement of PUE composites largely depends on the dispersion, special microstructures of nanofillers, and interface compatibility in the polymer matrix. Figure 8a_1_,a_2_ exhibits SEM images of neat PUE fracture surface with obvious perforation and a hole due to stress concentration of high-speed impact. Surrounded by a hole, there are many wrinkles and cracks. Figure 8b_1_,b_2_ presents a large amount of CNFs agglomeration behaviors and disorganized distribution in PUE matrixes, while CPC/PUE-Ⅴ have a better dispersion and compatibility as shown in Figure 8c_1_,c_2_, which is beneficial to stress transmission and energy absorption, and it can explain the reason why CPC/PUE had an excellent impact resistance under extreme conditions. What is more, CNFs are surrounded by CC-PEI, which brings in many active reactive groups, such as –OH, –NH/NH_2_, causing robust chemical bonding with the PUE matrix. In addition, there are plenty of π-π conjunctions between the CC-PEI@CNFs, which can provide flexible bonding to move in PUE matrix to achieve enough toughness to make impact stress transfer easily and absorb the impact energy, as shown in Figure 8d.

These enhancement effects can be ascribed to the CC-PEI co-deposition on CNFs, which builds special structures and improves the interface interaction and compatibility between CPC and PUE, making the stress transfer easier. The surface of the samples has no serious deformation or damage, while the neat PUE and CNFs/PUE show obviously irreversible deformation and perforation damage.

In all, the mass rate of CPC is one of the significant influence factors, which affects the dynamic impact resistance of PUE composites. Besides, compared with CNFs/PUE, the microstructures of PUE composites, which are designed as microscopic bone-like structures, have stronger impact resistance. These data indicate that CC-PEI@CNFs exhibit the significant role of preventing crack growth via bridging effect or inducing the crack deflection in the PUE matrix.

These experiments confirm that the PUE composites have the ability to resist permanent damage even under high strain rate load impact, revealing their potential for robust protection engineering.

## 4. Conclusions

In this work, inspired by the excellent impact resistance of human shin bones, we used CNFs and CC-PEI to form the microstructures of shin bones and studied the mechanical properties and dynamic response characteristics of PUE composites. CNFs are deemed to be tiny bones, which provide strength and rigidity, and CC-PEI can be served as “collagen”, providing robust adhesion and toughness, which are prepared by π-π conjugation and hydrogen bond, etc., noncovalent bonding and robust covalent bonding, to improve the dispersity and interface interaction between CNFs and PUE. The special structures of CC-PEI@CNFs can endow PUE with comprehensive mechanical properties and marvelous impact resistance, which are beneficial for the dynamic protection field.

## Figures and Tables

**Figure 1 nanomaterials-12-03830-f001:**
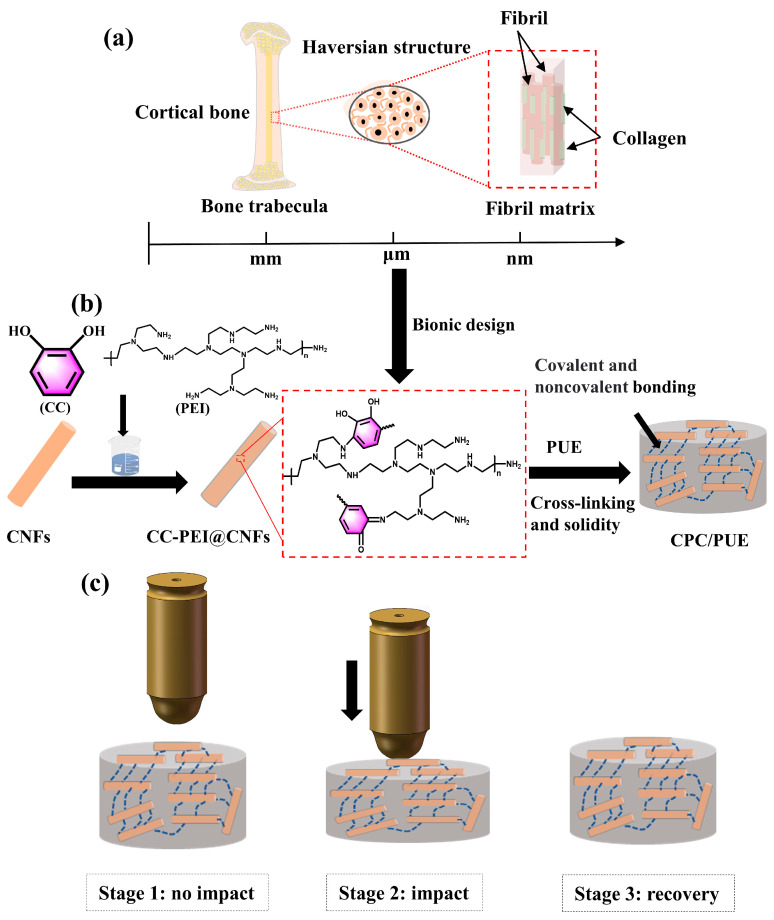
(**a**) Macro and micro structures of human shin bones. (**b**) Synthesis of PUE composites. (**c**) Excellent impact resistance of PUE nanocomposites with bone-like microstructures.

**Figure 2 nanomaterials-12-03830-f002:**
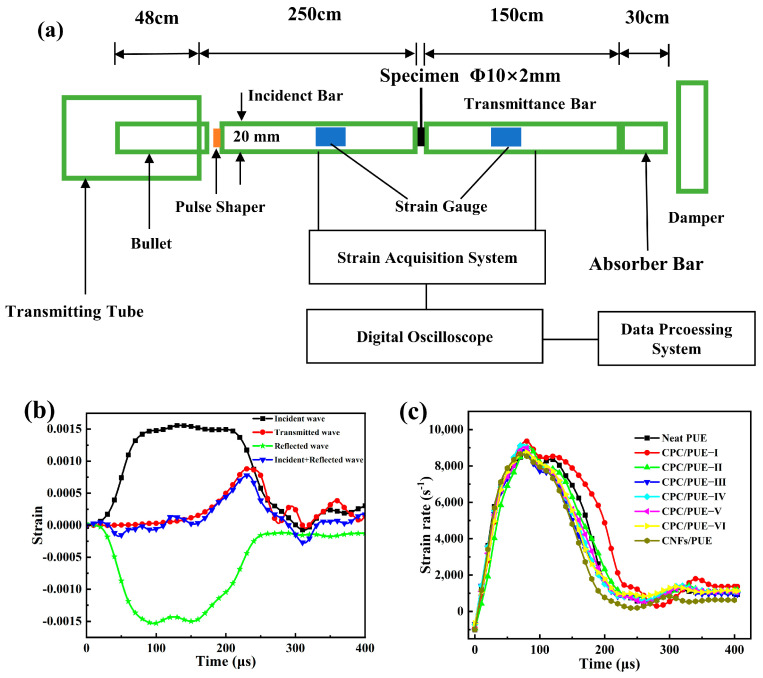
(**a**) SHPB apparatus schematic diagram. (**b**) Original waveform of SHPB test. (**c**) The strain rate of PUE composites after SHPB test.

**Figure 3 nanomaterials-12-03830-f003:**
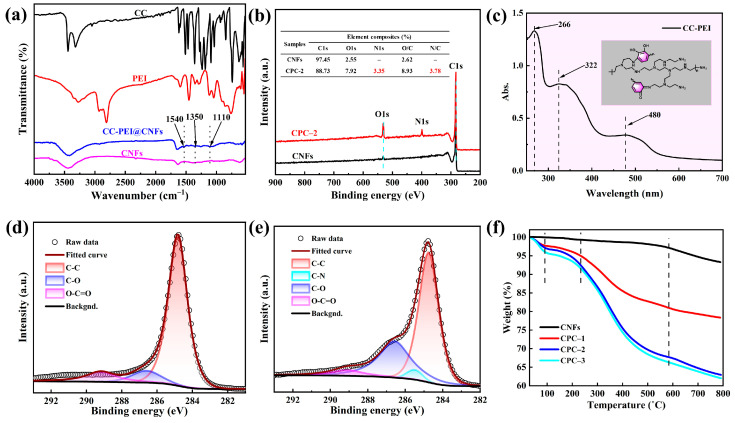
(**a**) FTIR spectra of CC, PEI, CNFs and CC-PEI@CNFs. (**b**) XPS full-scan spectra and element composition of the CNFs and CPC-2 (**c**) UV-Vis spectra of CC-PEI solutions after 36-h reaction. C1s peak fitting curves of (**d**) CNFs, and (**e**) CPC–2, (**f**) TGA spectra of CNFs, and different concentration gradient of CC-PEI to modify CNFs.

**Figure 4 nanomaterials-12-03830-f004:**
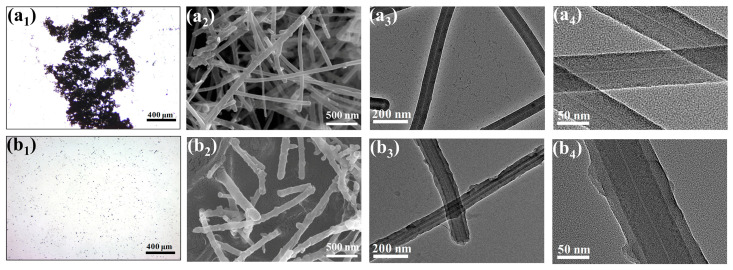
Optical microscope images of (**a_1_**) CNFs, (**b_1_**) CPC–2. SEM image of (**a_2_**) CNFs, (**b_2_**) CPC–2, TEM images of CNFs of (**a_3_**,**a_4_**), TEM images of CPC–2 of (**b_3_**,**b_4_**).

**Figure 5 nanomaterials-12-03830-f005:**
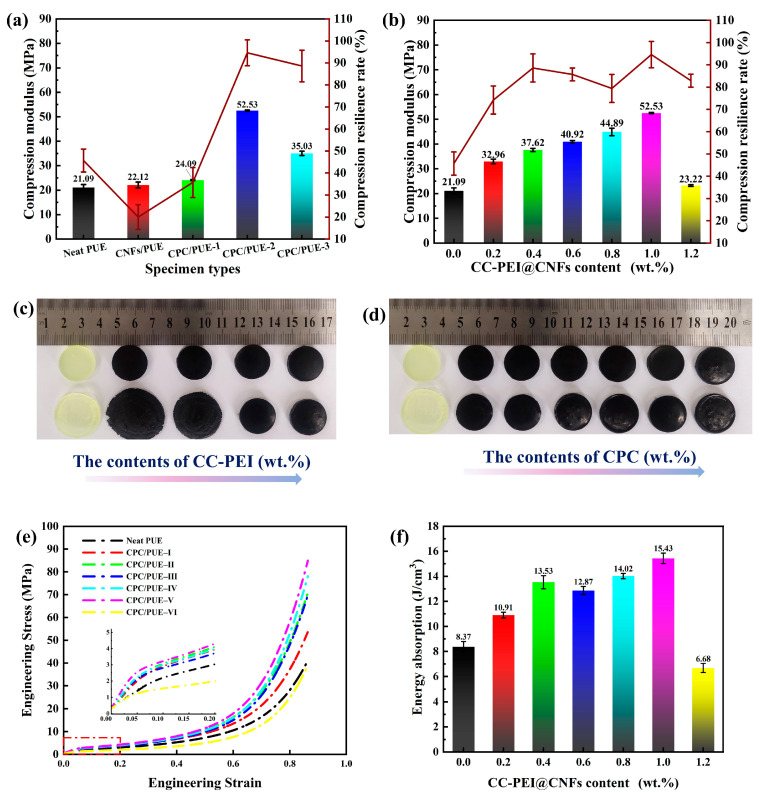
Compression modulus and compression resilience rate of PUE composites (**a**) at different concentrations of CC-PEI, (**b**) at different content of CC-PEI@CNFs, (**c**) experimental graphs of PUE composites at different concentrations of CC-PEI, (**d**) experimental graphs of PUE composites at different content of CC-PEI@CNFs, (**e**) stress–strain curves of PUE composites at the concentration of CPC-2 and magnified area of the top 0.2 strain, (**f**) static absorption energy of PUE composites.

**Figure 6 nanomaterials-12-03830-f006:**
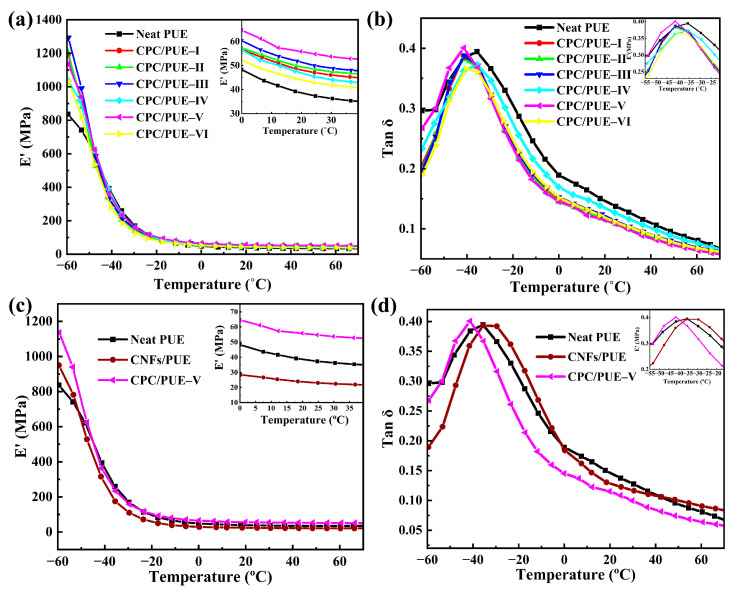
(**a**) E’, (**b**) tan δ and alterations as a function of temperature for neat PUE and CPC/PUE. (**c**) E’, and (**d**) tan δ and alterations as a function of temperature for the neat, CNFs/PUE, and CPC/PUE–Ⅴ.

**Figure 7 nanomaterials-12-03830-f007:**
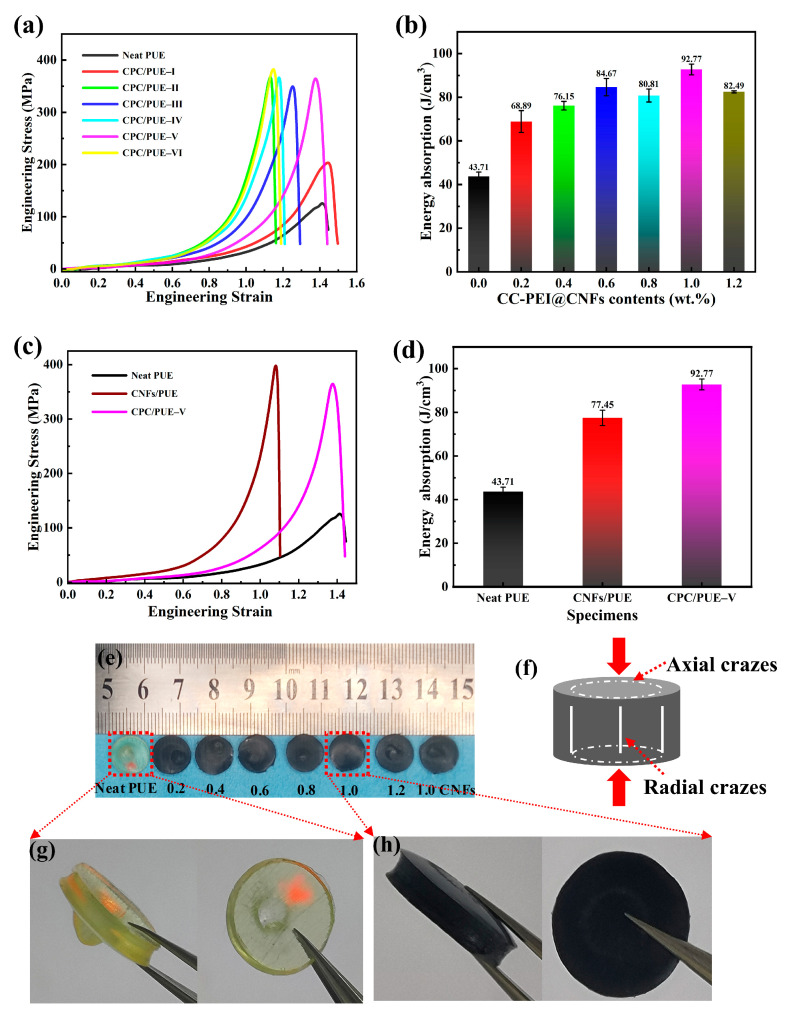
(**a**) Stress-strain curves of PUE composites tested by SHPB, (**b**) energy absorption of PUE composites, (**c**) stress-strain curves of the neat PUE, 1.0 wt.% CNFs/PUE and CPC/PUE-Ⅴ, (**d**) energy absorption capacity of the neat PUE, 1.0 wt.%CNFs/PUE and CPC/PUE-Ⅴ, (**e**) pictures of PUE and CPC/PUE after SHPB test. (**f**) Axial and radial failure of samples under high-speed impact. (**g**) Enlarged images of the neat PUE. (**h**) Enlarged images of CPC/PUE-Ⅴ.

**Figure 8 nanomaterials-12-03830-f008:**
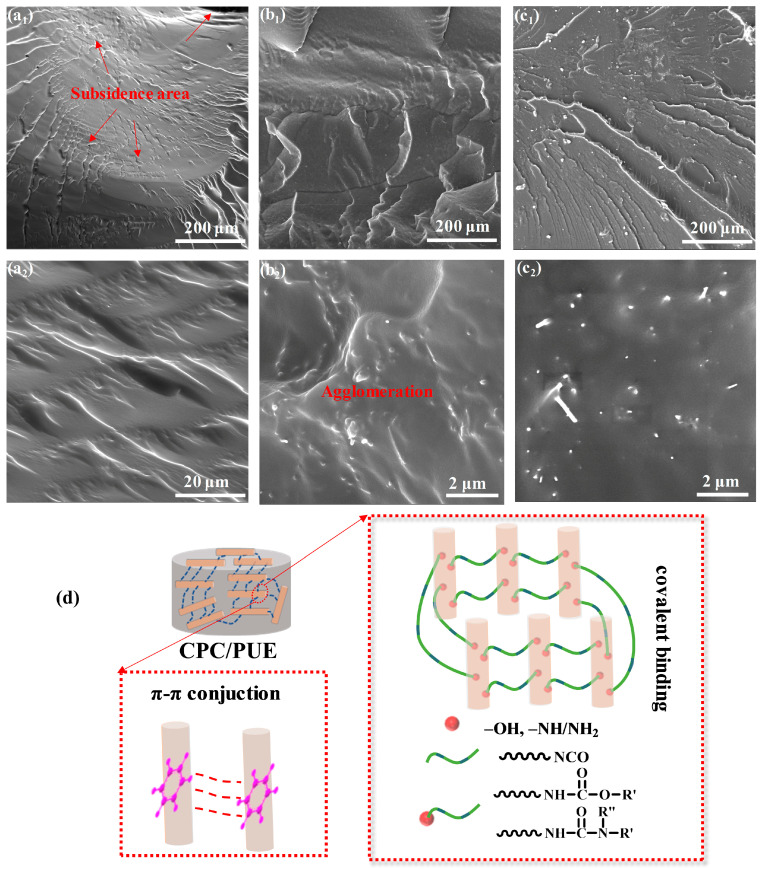
The SEM surface microtopography of the PUE sectional views of (**a_1_**,**a_2_**) the neat PUE, (**b_1_**,**b_2_**) CNFs/PUE, and (**c_1_**,**c_2_**)CPC/PUE–Ⅴ after SHPB test via liquid nitrogen quenching. (**d**) Schematic illustration of the excellent impact resistance of CPC/PUE.

**Table 1 nanomaterials-12-03830-t001:** Formulations of CC-PEI@CNFs (CPC).

Specimens	Contents of Materials (g)		
	PEI	CC	CNFs
CPC–1	0.20	0.10	0.70
CPC–2	0.50	0.25	0.70
CPC–3	1.00	0.50	0.70

**Table 2 nanomaterials-12-03830-t002:** The contents of the PUE composites.

Specimens	Contents of Materials (g)			
	CC-PEI@CNFs	CNFs	A Component	B Component
Neat PUE	–	–	10.000	4.000
CNFs/PUE	–	0.140	10.000	4.000
CPC/PUE–1	0.140	–	10.000	4.000
CPC/PUE–2	0.140	–	10.000	4.000
CPC/PUE–3	0.140	–	10.000	4.000
CPC/PUE–Ⅰ	0.028	–	10.000	4.000
CPC/PUE–Ⅱ	0.056	–	10.000	4.000
CPC/PUE–Ⅲ	0.084	–	10.000	4.000
CPC/PUE–Ⅳ	0.112	–	10.000	4.000
CPC/PUE–Ⅴ	0.140	–	10.000	4.000
CPC/PUE–Ⅵ	0.168	–	10.000	4.000

**Table 3 nanomaterials-12-03830-t003:** SEM line scanning of CNFs and CPC–2.

Specimens	C (at. %)	O (at. %)	N (at. %)
CNFs	93.00	7.00	0.00
CPC–2	80.90	11.00	8.10

**Table 4 nanomaterials-12-03830-t004:** Dynamic mechanical properties of neat PUE and its nanocomposites.

Specimen	E’@ −60 °C	E’@ 20 °C	T_g,s_ @ max. Tan δ (°C)	Tan δ Peak
Neat PUE	837.19 ± 79.89	38.47 ± 2.87	−35.64 ± 0.06	0.3906 ± 0.0016
CPC/PUE-Ⅰ	1189.81 ± 182.36	47.83 ± 2.26	−41.41 ± 0.26	0.3870 ± 0.0022
CPC/PUE-Ⅱ	1207.07 ± 28.71	49.10 ± 1.14	−41.73 ± 0.25	0.3820 ± 0.0005
CPC/PUE-Ⅲ	1293.42 ± 69.23	50.89 ± 3.26	−41.16 ± 0.29	0.3878 ± 0.0073
CPC/PUE-Ⅳ	1034.84 ± 1.49	46.52 ± 1.69	−39.93 ± 1.10	0.3730 ± 0.0032
CPC/PUE-Ⅴ	1135.80 ± 94.42	55.29 ± 1.09	−41.34 ± 0.12	0.4006 ± 0.0154
CPC/PUE-Ⅵ	1031.84 ± 23.25	43.75 ± 0.56	−40.33 ± 0.32	0.3665 ± 0.0020
CNFs/PUE	1256.89 ± 57.95	31.62 ± 2.41	−29.49 ± 0.03	0.3824 ± 0.0382

## Data Availability

Our study did not report any data.

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
