# Peer review of "Study on Bone-like Microstructure Design of Carbon Nanofibers/Polyurethane Composites with Excellent Impact Resistance"

_nanomaterials, 2022, doi:10.3390/nano12213830_

Round 1
Reviewer 1 Report
The article presented for review is very interesting, however, before the final publication, the authors should make some minor changes:
1. Authors should consider the naming of samples. At present, due to the large number of symbols, it is difficult to find out what material in the text is about.
2. Page 7. Verse 233. Instead of Fig5a it should be Fig5e.
3. Page 8. Verse 262. What is this "obvious regularity". Please explain.
4. Page 8. Verse 262. "Tg peak value slightly decreases" I disagree with this statement. The values are almost the same, and certainly much similar to each other.
Reviewer 2 Report
In my opinion, this is a very interesting work, with complete analysis and results.
Only to clarify some points:
· Please, provided error/standard deviation in results/tables.
· Specify if the curves obtained are representative curves of any case,
· Fig1. Please, check that the fig. caption descriptions according to all the subfigures are correct. All Figures/subfigures (i.e. Fig1 c,d,e) must be referenced in the text (in other case, they could be omitted)
· Fig5. Please, check that the fig. caption descriptions according to all the subfigures are correct.
· SPHB: which impact velocity was achieved?
Reviewer 3 Report
The special structures which are 22 combined of hard and soft, have a positive dispersion and compatibility in PUE matrix, which can 23 prevent cracks propagation by bridging effect or inducing the crack deflection. These PUE compo- 24 sites showed up to 112.26% higher impact absorbed energy and 198.43% greater dynamic impact 25 strength when compared to the neat PUE. The manuscript contains useful results and it is worth for publication after minor modification.
1. For XPS measuremnts the reference BE should be mentioned.
2. The structure characterization is valuable but several refences should be indicated. 400. 1 eV for nitrogen could be NH containing species (Top. Catal. 2018, 61, 1263-1273).
3. Several carbon containing species determined by XPS should be confirmed with literatures.
4. In comparison to the agglomeration of CNFs, the addition of CC-PEI on 213 the CNFs could observably improve the dispersity of nanofillers in the ethanol solution. Which properties of ethanol play a role? Polarity?
